# Exploring the Applications of Indocyanine Green in Robot-Assisted Urological Surgery: A Comprehensive Review of Fluorescence-Guided Techniques

**DOI:** 10.3390/s23125497

**Published:** 2023-06-11

**Authors:** Leslie Claire Licari, Eugenio Bologna, Flavia Proietti, Rocco Simone Flammia, Alfredo Maria Bove, Simone D’annunzio, Gabriele Tuderti, Costantino Leonardo

**Affiliations:** 1Urology Unit, Department of Maternal-Child and Urological Sciences, “Sapienza” University of Rome, Policlinico Umberto I Hospital, Viale del Policlinico 155, 00161 Rome, Italy; leslieclaire.licari@uniroma1.it (L.C.L.); roccosimone92@gmail.com (R.S.F.); 2Department of Urology, IRCCS “Regina Elena” National Cancer Institute, Via Elio Chianesi 53, 00144 Rome, Italy; flavia.proietti@uniroma1.it (F.P.); alfredo.bove@ifo.it (A.M.B.); simone.dannunzio@ifo.it (S.D.); gabriele.tuderti@ifo.it (G.T.); costantino.leonardo@uniroma1.it (C.L.)

**Keywords:** indocyanine green, ICG, fluorescence imaging, near-infrared fluorescence, NIRF, firefly, robotic surgery, urology

## Abstract

This comprehensive review aims to explore the applications of indocyanine green (ICG) in robot-assisted urological surgery through a detailed examination of fluorescence-guided techniques. An extensive literature search was conducted in PubMed/MEDLINE, EMBASE and Scopus, using keywords such as “indocyanine green,” “ICG”, “NIRF”, “Near Infrared Fluorescence”, “robot-assisted”, and “urology”. Additional suitable articles were collected by manually cross-referencing the bibliography of previously selected papers. The integration of the Firefly^®^ technology in the Da Vinci^®^ robotic system has opened new avenues for the advancement and exploration of different urological procedures. ICG is a fluorophore widely used in near-infrared fluorescence-guided techniques. The synergistic combination of intraoperative support, safety profiles and widespread availability comprises an additional asset that empowers ICG-guided robotic surgery. This overview of the current state of the art illustrates the potential advantages and broad applications of combining ICG-fluorescence guidance with robotic-assisted urological surgery.

## 1. Introduction

Over the years, technology has significantly changed surgical procedures.

Fluorescence-guided surgery (FGS) was first developed in 1947 with the introduction of fluorescein, a near-infrared fluorescent (NIRF) dye, for brain tumor resections [1]. Since then, many fluorescence-based techniques and fluorescent probes have been developed and used in different clinical scenarios [2]. Offering real-time feedback and supporting surgeons’ intraoperative decision-making, FGS has consolidated its role in changing surgical procedures.

In the past two decades, robot-assisted surgery (RAS) has revolutionized medicine prospects, allowing greater surgical precision along with a magnified view of tissues and anatomical structures [3,4,5].

Despite different clinical and surgical impacts, RAS and FGS, during their development, shared the same goals: improving surgeons’ accuracy, encouraging new surgical applications, and providing better possibilities in patients’ treatment. Since 2010, fluorescence imaging has been integrated into the Da Vinci^®^ Robotic System. The Firefly^®^ technology allows surgeons to switch vision modality from normal light to near-infrared light at any time during procedures, combining their respective benefits and improving the surgical experience.

Robotic surgery and fluorescence applications perfectly match the extreme heterogeneity offered by urological surgery.

## 2. Indocyanine Green Overview

Indocyanine Green (ICG) is a water-soluble fluorophore widely used in clinical research since its approval for intravenous (i.v.) administration by the FDA in 1956 [6]. Its green fluorescence, emitted when excited by near-infrared light, can be detected using dedicated optical systems without affecting the surgical field view [7,8,9].

After i.v. administration, ICG rapidly binds to serum proteins, confining it to the vascular compartments [10].

Characterized by a plasma life of 3–5 min and exclusive biliary excretion within 10–15 min, ICG shows a high-safety index and nonradioactive properties [11].

The recommended safe dose for a standard diagnostic procedure is 0.1–0.5 mg/kg [12]. If injected directly into tissues, ICG migrates in lymphatic vessels and in lymph nodes, where it deposits into macrophages [13] and can provide information about organs’ lymphatic drainage.

Its properties make it useful for the visualization of vascular anatomy, the assessment of tissue perfusion, lesion “tattooing” [14] and lymph node road mapping [15]. Urological surgery is one area where ICG’s versatility and safety profile have made it increasingly popular.

## 3. NIRF-Guided Robot-Assisted Renal Surgery

The use of Indocyanine Green in nephron-sparing robotic surgery has gained widespread recognition as a versatile, valuable, and safe technique since its introduction about 15 years ago.

With its support in delineating tumor margins and vascular anatomy, in guiding selective or super selective clamping and in assessing kidney perfusion after resection and renorrhaphy, ICG offers real-time guidance to promote the better preservation of renal function (Table 1).

### 3.1. ICG-Guided Renal Mass Differential Fluorescence

The idea behind the use of ICG as a “tumor marker” in kidney cancer surgery is based on experimental evidence. Cortical tumors show a downregulation of bilitranslocase, a carrier protein that allows ICG uptake into the cells [16], resulting in a fluorescence-based visual differentiation from the normal surrounding parenchyma.

Tobis et al. were the first, in 2011 [17], to evaluate the additional value offered by ICG-fluorescence in 11 cases of robot-assisted partial nephrectomy (RAPN). The goal was to utilize the i.v. injection of ICG to differentiate between normal parenchyma and malignant tissue and to highlight the renal vasculature. ICG-guided imaging was deemed potentially useful in eight patients for outlining the resection margin, and in all patients for vessel identification.

The following year, the same group speculated on the potential role of ICG in distinguishing between benign lesions (iso- or hyperfluorescent) and malignant lesions (hypofluorescent) [18], but the results were not supported by subsequent studies [19,20].

What emerged as a possible watershed for the effectiveness of this technique was the ICG dosage. ICG under-dosing “tones down” the normal parenchyma fluorescence, leading to uncertainty in distinguishing between tumor and healthy tissue. On the other hand, ICG overdosing “tones up” all the renal parenchyma, leading to an opposite situation with the same results. Surgeons who have abandoned the use of NIRF or have reported a lack of effectiveness for differential fluorescence and visual margin assessment have probably used excessively high doses of ICG [21].

In 2013, Angell et al. [22] reported a reproducible scheme to achieve a correct ICG dose. They described a first dosage as a test, followed by a second dosage—calibrated on the first one—to achieve the right contrast, reporting a successfully differential fluorescence in 65 out of 79 tumors.

Similar results came from a more recent prospectively collected database of 325 patients who underwent ICG-RAPN [21]. Using the same concept—an initial low-test dose and an optional second dose immediately prior to vessels clamping—the authors reported an overall success in differentiation rate of 87.3% and an extremely low positive surgical margin rate (0.30%).

### 3.2. ICG-Guided Renal Selective Perfusion Assessment

To reduce ischemic injury during partial nephrectomy, Gill et al. [23] introduced in 2011 the concept of “zero-ischemia”. An anatomic microdissection of the renal vessels allows a selective or super selective clamping of the tumor-feeding artery, avoiding global ischemia.

The angiographic properties of ICG showed to be very useful in this setting; ICG imaging can confirm tumor devascularization and normal kidney perfusion and can even identify missed arterial branches if the expected regional perfusion deficit is not reached. This allows surgeons to adjust clamping, reduce bleeding and improve tumor excision quality.

Many studies indeed reported that “zero-ischemia” RAPN with NIRF was a safe alternative to conventional on-clamp RAPN, and may improve functional short-term outcomes [24,25,26,27,28]. A statistically significant benefit of a eGFR variation at discharge was reported, supporting the ICG-zero ischemia approach (ΔeGFR ≈ 15%) [24,25,28,29].

A subsequent meta-analysis [30] revealed no significant difference between on-clamp and selective clamping techniques in terms of complications, positive surgical margins (PSM), operative time and estimated blood loss.

In 2020, in a large multi-institutional study of 737 patients, Diana et al. [31] tried to define the role of fluorescence-guided surgery during RAPN. They performed a subgroup analysis comparing ICG-RAPN with standard RAPN. According to the authors’ suggestions, ICG showed a potential advantage for challenging vascular anatomy—such as large tumors with more complex and accessory vascularization and horseshoe, solitary or pelvic kidney—or impaired renal function. However, despite a correlation between ICG and trifecta achievement, the study described no clear functional advantages.

In 2022, Yang et al. [32] described a short-term advantage in preserving eGFR in the ICG-RAPN group, which decreases over time (3 months vs. 6 months). The authors suggested that the standard-RAPN group may be more susceptible to acute tubular necrosis due to increased renal ischemia. This results in an initially poorer eGFR preservation, which gradually rose during the recovery phase. Gradual compensation by the normal contralateral kidney may also minimize the long-term advantage of the ICG-group.

This finding was recently confirmed by the EMERALD randomized single-blind trial [33]. The study was interrupted after 30 cases; at 6 months follow-up, the interim analysis showed no benefit in the ICG super selective RAPN approach evaluated as a combination of eGFR and relative renal function on ^99m^Tc-DMSA scintigraphy (−21.4% vs. −23.4%, *p* = 0.66). This lack of difference persisted even after adjusting for the percentage of kidney volume preserved—an independent predictor of functional preservation. This trial raises further questions regarding the usefulness of this technique that underlies an intrinsic increased technical risk of vascular damage.

In any case, other studies like the one conducted in 2023 by De Backer [34] highlight how ICG can be used to support the development of new technologies in the field of nephron sparing surgery. Their study utilized ICG administration as a tool to validate their perfusion zone algorithm integrated into a 3D model for planning super selective clamping.

### 3.3. ICG-Guided Management of Endophytic Renal Tumors

Endophytic renal masses represent a surgical challenge during nephron-sparing surgery due to technical difficulties and a higher risk of complications [35] that potentially narrows the indication for conservative surgery [36]. The standard intravenous ICG use has played a marginal role in this setting because the tumor location reduces the benefits offered by FGS [19].

An innovative application of indocyanine green in this field has been proposed by Simone et al. in a series of 10 totally endophytic renal masses. The patients received a preoperative super selective trans-arterial delivery of an ICG-Lipiodol mixture into tertiary-order arterial branches—feeding the renal mass—prior to transperitoneal “purely” off-clamp RAPN [37]. The ICG-marked area guided not only the tumor localization, but also a safer enucleation. The authors reported several benefits, including preoperative resection strategy improvement, quick intraoperative mass identification, and real-time control of resection margins. The feasibility of this “tattooing” technique, even with different ICG-mixed agents, has been confirmed by subsequent studies [14,38].

In 2022, Nardis and colleagues evaluated the clinical impact of ICG combined with Lipiodol in the context of trans-arterial super selective embolization (Figure 1), in a cohort of 41 patients with totally endophytic masses [39]. The study reported a procedure success rate of 100%, and 63.4% of the tumors were considered “visible with well-defined margins” intra-operatively. Combining ICG with other emerging technological tools for preoperative surgical strategies, such as three-dimensional (3D) reconstructions, may further enhance the potential of RAPN. An example of this application was illustrated in a pilot study by Amparore et al. published in 2023 [40], where ICG was used to overlay a 3D virtual model of the kidney onto the real organ during surgery, with promising preliminary results.

## 4. NIRF-Guided Robot-Assisted Renal Transplant

Rosales et al. [41] were the first to report their experience in kidney transplantation with a pure laparoscopic approach, introducing the kidney through a Pfannenstiel incision. The value of robotic-assisted kidney transplant (RAKT) was not recognized until 2014, when the Vattikuti Urology Institute-Medanta collaboration introduced the novel surgical technique for RAKT [42] following a safer step-by-step process to introduce surgical innovations into real-life practice [43]. A multicenter prospective study has subsequently confirmed the safety and feasibility of RAKT [44].

Following the evolution of kidney transplant surgery, ICG has not lost its pace and has promptly been incorporated into this technological process, maintaining the advantages expressed during conventional surgery, especially regarding the assessment of perfusion and anastomosis quality evaluation [45] (Table 1).

In 2019, Vignolini and colleagues [46] reported their preliminary experience of an intraoperative objective assessment of graft reperfusion with ICG during RAKT. After the completion of vascular anastomoses, 0.3 mg/kg of ICG was injected intravenously, obtaining a quantitative intraoperative feedback of graft reperfusion. In addition, before uretero-vesical anastomosis, ICG allowed the surgeon to have a real-time assessment of ureteral vascularization, granting the possibility to tailor the ureteral length based on the fluorescence signal. In recent years, ICG continues to offer new applications in the field of kidney transplants. An example of this novelty is represented by the prospective study published in 2023 by the Italian group of Ietto and colleagues [47]. They proposed an additional evaluation—along with pretransplant biopsy findings and renal resistive index—of kidney vascularization before the implant. After injecting 5 mg of ICG directly into the renal artery, they assessed the status of kidney microcirculation through fluorescence intensity. The authors, evaluating the relationship between the mean fluorescence intensity and the graft’s outcomes, concluded that the quantitative assessment of fluorescence before grafting—combined with the donor’s medical history and biopsy scoring—could guide the physician in graft quality evaluation.

**Table 1 sensors-23-05497-t001:** Summary of indocyanine green (ICG) applications in robotic renal surgery.

Robotic Procedure	Purpose	Potential Pros	Potential Limitations	ICG Administration
Partial Nephrectomy	Differential fluorescence to assess tumor margins [17,18,19,20,21,22]	Real-time guidance Maximal preservation of renal parenchyma	Doses of ICG outside an optimal range result in decreased contrast between the lesion and surrounding renal parenchyma Limited tissue penetration	Intravenous injection prior to arterial clamping
	Perfusion assessment for selective arterial clamping or test clamping of main artery [24,25,26,27,28,29,30,31,32,33]	Useful in cases of challenging vascular anatomy or impaired renal function Monitoring segmental perfusion deficits after clamping	Limited assessment of deep devascularization	Intravenous injection after arterial clamping
	Assess kidney perfusion after resection and renorrhaphy [24,26,31,32,37]	Checking residual parenchyma blood supply and confirming absence of ischemic injury to healthy parenchyma	Lack of data about specific decision making and clinical impact	Intravenous injection after reperfusion
	Intraoperative identification and anatomical dissection of total endophytic renal masses [14,37,38,39]	Real-time guidance May improve preoperative resection strategy and intraoperative mass identification May promote nephron-sparing surgery	Needs preoperative renal mass marking No free dye application No benefit in case of avascular renal masses	Preoperative superselective catheterization of tertiary arterial branches feeding the tumor by interventional uroradiologist +\− embolization
Renal Transplant	Assessment of graft perfusion before and after transplant [46,47]	Depicting graft microcirculation Evaluating ureteral reperfusion Useful for complex vasculature reconstruction	Preliminary experience No long-term outcomes evaluation	Intravenous injection Renal artery injection before the implantation

## 5. NIRF-Guided Robot-Assisted Adrenal Surgery

The promising results of ICG-fluorescence in adrenal laparoscopic surgery [48] were promptly also confirmed in the robot-assisted approach, offering new opportunities for adrenal-sparing procedures in selected patients (Table 2).

Manny and colleagues [49] were the first to report robot-assisted partial adrenalectomy (RAPA) guided by ICG in patients with small adrenal masses. In a total of three patients, ICG fluorescence helped to discriminate the lesion from normal parenchyma—based on fluorescence intensity—allowing a safe adrenal-sparing approach with complete excision.

In 2015, in a feasibility study by Sound et al. [50], ICG use was described in 10 patients undergoing robotic radical adrenalectomy (RAA). The study confirmed the utility of an intravenous ICG low dose (3–3.5 mg) to identify the adrenal gland, guiding the dissection.

Gokceimam and colleagues [51] reported similar findings in a series of robotic posterior retroperitoneal adrenalectomies in 2021. The fluorescence delineation between adrenal gland and retroperitoneal tissues reached its peak 5 min after injection—with multiple injections required—and persisted for up to 20 min.

A further larger study conducted by Colvin et al. [52] attempted to highlight the real added value offered by ICG during intraoperative dissection. A total of 40 patients underwent 43 adrenalectomies with both lateral transabdominal and posterior retroperitoneal approaches. ICG imaging was superior, equivalent or inferior to the conventional robotic view—based on independent study reviewers’ assessments—in 46.5%, 25.6% and 27.9% of the procedures, respectively. Tumor type was the only parameter that predicted the superiority of ICG imaging. Specifically, adrenocortical tumors offered fluorescence properties that outperformed the conventional robotic view.

Kahramangil and collogues, in 2018 [53], published their results from 100 procedures, confirming a different fluorescence pattern among adrenal masses according to histologic origin. ICG conferred the highest utility for adrenocortical tumors and cortical-preserving adrenalectomy for pheochromocytoma, offering a superior margin distinction compared to the non-fluorescent view. Tumors of different origin, conversely, did not prove the same advantages.

Recently, the study by Aydin et al. [54] succeeded in quantifying the subjective benefits of NIRF-ICG in RAA, using color analysis. The researchers used a visual grading scale to assess the quality of tissue visualization, with higher scores indicating better tissue distinction. The results showed that the use of ICG significantly improved tissue visualization and distinction, compared to conventional views.

Although no studies have been specifically designed to assess whether ICG guidance decreases complications, existing data suggest that its use can improve the speed and quality of dissection by reducing the risk of bleeding and capsular violation. Additionally, ICG can help identify the adrenal vein—as it remains hypofluorescent—which could improve surgical accuracy [54].

**Table 2 sensors-23-05497-t002:** Summary of indocyanine green (ICG) applications in robotic adrenal surgery.

Robotic Procedure	Purpose	Potential Pros	Potential Limitations	ICG Administration
Partial and Radical Adrenalectomy	Tissue identification and dissection Identification of vasculature anatomy [49,50,51,52,53,54]	Better delineation between adrenal gland and retroperitoneal tissues	Multiple injections required Background liver fluorescence can interfere during right-sided posterior retroperitoneal approach	Intravenous injection
	Tumor localization [49,52,53]	May promote adrenal-sparing surgery	Fluorescence pattern variability based on histological tumor origin	Intravenous injection

## 6. NIRF-Guided Robot-Assisted Prostate Surgery

### 6.1. ICG-Guided Nerve-Sparing Approach during RARP

An increased understanding of the prostatic neurovascular bundle (NVB) anatomy has led various authors to describe different nerve-sparing approaches to maximize functional outcomes during robot-assisted radical prostatectomy (RARP) [4,55]. Researchers have defined the potential role of a “landmark” artery—a network of capsular arteries running along the lateral border of the prostate related intimately to the cavernosal nerves—to better delineate the NVB [56].

In 2015, Kumar and colleagues evaluated ICG fluorescence in supporting the identification of this “landmark” during RARP (Table 3). In a total of 20 neurovascular bundles evaluated—after i.v. injection of ICG—the authors reported an identification rate of 85%, potentially associated with nerve-sparing quality improvement [57]. Better results have been described by Mangano et al., who reported the identification of NVB in 100% of patients without increasing the operative time [58].

More recently, in 2023, Amara et al. [59] published a retrospective study on a total of 91 RARP. According to the authors’ perceptions, fluorescence provided not only a real-time visualization of the NVB, but even a better understanding of its “over traction” during dissection. In their cases, the use of ICG led to erectile function recovery in most patients at 9 months without increasing the operative time or the risk of complications.

### 6.2. ICG-Guided Lymphadenectomy during RARP

Despite the progress made by new molecular imaging techniques [60], extended bilateral pelvic lymphadenectomy (ePLND) remains the best approach for nodal staging in intermediate and high-risk prostate cancer (PCa) patients with a nomogram-assessed risk greater than 5% [61]. Over time, the utilization of tracers such as ICG, linked with the sentinel lymph node (SLNs) concept, has been focused towards two principal objectives. The primary objective, albeit carrying an increased risk of morbidity, was centered on maximizing the number of lymph nodes obtained during ePLND to bolster the capacity of tumor staging. The second objective, conversely, was centered on a targeted nodes dissection with a higher probability of harboring malignancy.

Manny and colleagues [62], in 2013, published the first series of 50 consecutive RARP with ICG-lymphography. ICG solution was injected into each prostatic lobe using a robotically guided percutaneous needle. Fluorescence was able to identify the potential sentinel prostatic drainage in 76% of patients, with a sensitivity and specificity of 100% and 75.4%, respectively.

These promising results were not confirmed by a randomized trial published in 2018 evaluating the impact of prostatic ICG injection in a total of 120 patients [63]. Among the intervention group, ICG was applied transrectally, with two basal, two apical and one central injection. In this series, seven out of nine patients with pN1 disease presented at least one fluorescent lymph node, but 35 non-fluorescent metastases were missed by ICG-fluorescence, resulting in a sensitivity of 44%. In accordance with the authors’ hypothesis, a higher metastatic burden could obstruct the lymphatic pathways, leading to a disrupted passage of the tracer.

Further studies confirmed these findings [64,65]. In the recent study by Ozkan and collegues [65], among a total of 25 patients in the ICG-group, they reported 11 non-fluorescent metastases that were missed, and only 9 fluorescent cancerous nodes identified, resulting in a sensitivity rate of 45%.

Despite the limited usefulness of the free dye in identifying metastatic lymph nodes, ICG guidance can still provide advantages in performing high-quality ePLND. As suggested by Shimbo et al. [64], indocyanine green real-time guidance can assist inexperienced surgeons or institutions in identifying LN drainage pathways and residual lymph nodes for dissection, even outside standard templates (Table 3).

Van der Poel et al. took a step forward, describing the properties of ICG in conjunction with radioactive tracers [66]. They described a hybrid radiocolloid tracer (ICG-^99m^Tc-NanoColl) detectable using gamma camera, computed tomography (CT), and NIRF imaging. Mazzone and colleagues, in 2021, evaluated this additional role of combined fluorescence and radioactive tracers during SLNs biopsies among a large sample size of 1680 patients [67]. They concluded that the hybrid tracer improved the lymph node involvement detection rate by 10%, reducing the risk of false-negative findings, without increasing postoperative complications.

Recent research also suggests that variances in the intraprostatic injection site could affect the anatomical location, as well as the number of SLNs identified [68,69]. Based on this evidence, the aim of a recent phase II RCT conducted by Wit and colleagues [70] was to establish whether intratumoral injection (IT) improves the detection of positive sentinel lymph nodes compared to traditional peripheral zone (IP) injection.

IT, according to the study results, did not significantly affect the number of removed SNLs but provided a higher positive rate compared to IP injection. Thus, the authors support the idea that IT injection may be a viable option for detecting LN involvement, with a potentially higher positive detection rate, but the optimal balance between sensitivity and specificity would be achieved when IT- and IP-procedures are combined.

The same group published a phase II trial in 2023 evaluating the diagnostic accuracy of the hybrid tracer (ICG-^99m^Tc-nanocolloid) compared to a sequential use of ^99m^Tc-nanocolloid pre-operatively and an intraoperative ICG intra-prostatic injection [71]. The hybrid tracer, according to the study results, improved the positive predictive value for lymph node involvement, minimizing the number of fluorescent nodes compared to the sequential tracer approach.

### 6.3. ICG-Guided Urethra-Sparing Simple Prostatectomy

With the increasing adoption of robotic surgery in patients with BPH, robot-assisted simple prostatectomy (RASP) may be considered an alternative to the standard open technique, also offering interesting new perspectives regarding the use of ICG.

The preservation of the prostatic urethra (Madigan technique)—described by Dixon et al. in open surgery [72]—showed a significant advantage [73] but it has been progressively abandoned due to the significant complexity in avoiding the unintentional violation of urinary tract.

Simone and colleagues evaluated the possible added value offered by the intraurethral injection of ICG during the Madigan approach. In 12 consecutive patients who underwent RASP, an intra-urethral injection of ICG guided adenoma dissection, ensuring the selective control of the intraprostatic urethra and minimizing and clearly identifying any unintentional violation of the urinary system. The authors reported the feasibility of this technique, with anterograde ejaculation preserved in eight patients and no need for continuous bladder irrigation in 83.7% of patients [74] (Table 3).

## 7. NIRF-Guided Robot-Assisted Penile Cancer Surgery

Similar to what has been highlighted in prostate cancer, surgical lymph node staging continues to be the gold standard for clinically node-negative (cN0) patients with intermediate-risk and high-risk ≥pT1G2 disease [75].

Overtreatment occurs in approximately 80% of patients, carrying a higher risk of comorbidity [76,77], but the association between inguinal lymphadenectomy and improved overall survival has been demonstrated [78].

Over the years, ICG-guided lymphoadenectomy has demonstrated better properties—even in combination with radiotracers—compared to traditional blue dye. Specifically, ICG has been found to exhibit rapid dissemination through the lymphatic system and superior tissue penetration, without compromising the surgical field.

In 2014, a pilot study involving 65 patients with penile squamous cell carcinoma (SCCp) compared the use of a hybrid ICG-^99m^Tc-nanocolloid tracer to the standard ^99m^Tc-nanocolloid tracer with blue dye. Intraoperatively, ICG-fluorescence visualized a significantly higher number of sentinel nodes compared to the control group (96.8% versus 55.7%; *p* < 0.0001) [79]. These results, confirmed by further studies [80], advocated the added value of fluorescence to improve intraoperative sentinel node identification.

The subgroup analysis conducted by Dell’Oglio and colleagues [81] on 400 patients with SCCp provided supplementary data. Fluorescence sentinel node biopsy based on ICG-^99m^Tc-nanocolloid confirmed a better detection rate compared to hybrid-blue dye (95% vs. 56%, respectively), but it showed a slightly lower rate when compared to gamma-probe SNLs detection (96% vs. 98%, respectively). However, fluorescence enabled the identification of SNLs affected by radioactive decay—so not identifiable by the gamma probe—that consolidated the benefits offered by the combination of these two detection modalities [82].

A recent prospective study, conducted in 2022, underlined divergent results [83] among a total of 414 sentinel lymph nodes specimens. The authors reported a low identification rate based on fluorescence (85%) and a relatively high rate—compared to previous results—using blue dye (82%). This finding is potentially explained by differences in the tracer injection technique, patient characteristics, and perioperative conditions. An alternative interpretation, according to the authors’ suggestion, would be that with adequate technical skills and appropriate patient selection, these two different approaches may achieve similar abilities in target detection.

Only a few studies have reported on the effectiveness of using ICG guidance for robotic inguinal lymphadenectomy (RILND) in the treatment of penile cancer [84,85] (Table 3). Bjurlin et al. [86] and Savio et al. [84] demonstrated the safety and feasibility of this technique. They visualized the lymphatic channels and nodes using the Firefly^®^ technology after injecting ICG intradermally at the base of the penis or below the tumor, respectively. No postoperative complications were recorded in either study. In 2022, Yuan and colleagues [85] published a novel technique for bilateral RILND. Using a hypogastric subcutaneous approach, 10 patients were included in the ICG group and 16 patients were included in the non-ICG group. The numbers of retrieved bilateral superficial and deep inguinal LNDs were higher in the ICG group than in the non-ICG group (29.5 vs. 25, *p* < 0.001), with no significant differences between the two groups regarding postoperative complications.

**Table 3 sensors-23-05497-t003:** Summary of indocyanine green (ICG) applications in robotic prostatic and groin surgery.

Robotic Procedure	Purpose	Potential Pros	Potential Limitations	ICG Administration
Radical Prostatectomy	Identification of neurovascular bundle and landmark prostatic artery [57,58,59]	High identification rate May improve nerve-sparing quality	Only reports with low sample size Lack of evidence about postoperative functional outcome correlation	Intravenous injection
	Lymphangiography and lymphadenectomy [62,63,64,65,66,67,68,69,70,71]	Simpler nodes identification Higher number of LND yield Can give information about prostatic lymphatic drainage routes, also outside standard templates Can be used with radiotracers to improve accuracy	Low sensitivity and specificity in detecting metastatic LN Not an alternative to ePLND Variable drainage related to tumor burden Lack of a standard injection protocol	Intraprostatic/intratumoral injection
Simple Prostatectomy	Urethra identification and preservation [74]	Selective control of the intraprostatic urethra Identification of unintentional violation of the urinary system	Only exploratory report For urethra-sparing technique only	Retrograde injection through catheter
Inguinal lymphadenectomy for Penile cancer	Sentinel lymph node mapping [84,85,86]	May improve intraoperative optical SN detection rate	Only initial reports with low sample size Lack of a standard injection protocol	Intradermal injection at the base of the penis Subcutaneous injection below the tumor

## 8. NIRF-Guided Robot-Assisted Upper Urinary Tract Surgery

Fluorescence-guided surgery has proven to be a valuable tool in upper urinary tract surgery, where the precise identification and preservation of critical structures is essential for optimal patient outcomes (Table 4). The intraluminal injection of ICG—through nephrostomy or directly into ureters—allows for the real-time delineation of the ureter and renal pelvis [87]; the intravenous injection of ICG allows a safe reconstruction, assessing tissue vascularization during the reconstructive phase [88].

Lee et al. [87], in 2013, presented these advantages in seven robot-assisted ureteroureterostomy treatments for ureteral stenosis. According to the authors’ perception, the intraoperative instillation of ICG above and below the stenotic tract facilitated a better localization and delineation of stenosis’s characteristics and the distinction of a healthy ureter based on fluorescence intensity.

The same group confirmed [89] these advantages in other robot-assisted upper tract reconstructive techniques including ureterolysis, pyeloplasty, ureteroureterostomy and ureteroneocystostomy. ICG-fluorescence permitted a simpler identification of the ureter in all patients, a less unnecessary tissue disruption and a more precise localization of the proximal and distal stricture margins.

In 2014, Bjurlin and colleagues [88] presented additional possibilities offered by ICG during pyeloplasty and ureteroureterostomy. They hypothesized that the use of NIRF imaging has the potential to predict and prevent postoperative obstruction and pyeloplasty failure. Prior to performing the anastomosis, ICG is administered intravenously, then fluorescence intensity assesses structures’ perfusion, minimizing the potential risk connected to unhealthy tissue reconstruction. In a total of 47 procedures, the authors reported radiographic and symptomatic improvements in 100% of the pyeloplasty, ureteral reimplant and ureteroureterostomy patients and 71.4% of ureterolysis patients, for an overall success rate of 95.2%.

More recently, in 2023, Zeng et al. [90] published data of fourteen patients: three ureteral strictures, five ureteropelvic junction obstructions, four duplicate kidney and duplicate ureters, one megaureter and one ipsilateral native ureteral tumor after renal transplantation.

In this series, ICG facilitated ureter identification in complex anatomy (duplicate ureter or naïve ureteral tumor after transplantation); improved the tissue vitality assessment in reconstruction techniques (megaureter, UPJ obstruction and ureteral strictures); and ensured quality control after anastomosis or reimplantation, preventing leakage or re-stenosis.

## 9. NIRF-Guided Robot-Assisted Bladder Surgery

The role of NIRF imaging in bladder surgery is still under exploration. Several studies have evaluated the potential benefits offered by fluorescence during radical cystectomy, from a clearer identification of lymphatic drainage, including sentinel lymph node mapping and extended lymph node dissection tailoring, to an improved evaluation of tissue vitality during the reconstructive phase (Table 4). Lymph nodes’ status in bladder cancer significantly affects clinical decision-making, and various modalities for lymph node assessment have been investigated, including intraoperative ICG-fluorescence [91].

In 2012, Inoue and colleagues [92] were the first to report the safety and feasibility of lymph node mapping based on ICG fluorescence in a total of 12 patients undergoing radical cystectomy and pelvic lymphadenectomy. A 0.5-mL solution containing 0.25% ICG was injected into the bladder near the tumor intraoperatively. In seven patients, ICG allowed a clear identification of the lymphatic pathway, but it improved the detection of only non-metastatic nodes, which limited its usefulness.

Better results were reported by Schaafsma et al. [93] in 2014. They evaluated different injection techniques—cystoscopically around the tumor or from outside through the sierosa after laparotomy—of ICG combined with human serum albumin.

The authors reported a 92% SLN detection rate when ICG mucosal injection around the tumor was followed by saline bladder distension, pointing out how bladder filling was directly correlated with a better delineation of the lymphatic pathway.

Additionally, in 2014, Manny et al. published the results of the first application of an NIRF-guided sentinel lymph node biopsy in robot-assisted radical cystectomy (RARC) [94]. They administered submucosal and detrusor injections of ICG around the tumor and found that 90% of patients showed visible sentinel drainage with heterogeneous patterns. The study also revealed that ICG lymphangiography had a sensitivity of 75% in correctly identifying nodal metastasis, but a low specificity of 52%.

A subsequent comparison between fluorescent and radio-guided lymph node mapping in bladder cancer also demonstrated high sensitivity but a higher rate of false positives in the ICG group [95].

Since then, probably due to the low specificity provided by the free dye, the research on ICG applications in this field seems to have lost its momentum. Moreover, even the combination of pre- and intraoperative SN imaging and hybrid radiotracers has not yet created a breakthrough for MIBC. A recent study by Rietbergen et al. [96] combined the use of an ICG hybrid tracer (ICG-^99m^Tc-nanocolloid), preoperative lymphatic mapping through static lymphoscintigraphy and SPECT/CT and intraoperative radio- and fluoresce guidance (via gamma probe and fluorescence camera, respectively). Despite these measures, the authors reported an SN non-visualization rate of 47%, raising serious doubts regarding the added value offered by sentinel lymph node biopsy.

Conversely, the application of NIRF-ICG imaging for assessing bowel and distal ureters’ vascularization during urinary derivation after RARC has been found to be beneficial in reducing post-operative complications and improving outcomes. Several studies reported the usefulness of this application [97,98,99]. Ahmadi et al. [99] reported a statistically significant reduction in per-patient anastomotic stricture rate in the ICG group compared to the non-ICG group (0% vs. 10.6%, *p* = 0.020) after a median follow-up of 12 months.

Furthermore, ICG shows its efficacy not only in preventing complications, but also in the treatment setting [100,101]. Tuderti et al. [101] reported a series of robotic ureteroileal reimplantations (RUIR) for ureteroileal anastomosis strictures, where near-infrared fluorescence imaging was used after the intravenous and trans-nephrostomic anterograde injection of ICG (Figure 2). In these cases, NIRF imaging provided a useful guide for promptly identifying and progressively dissecting the ureter, thereby reducing the risk of intraoperative vascular and bowel injuries with a high success rate.

**Table 4 sensors-23-05497-t004:** Summary of indocyanine green (ICG) applications in robotic bladder and upper tract surgery.

Robotic Procedure	Purpose	Potential Pros	Potential Limitations	ICG Administration
Radical Cystectomy	Mesenteric angiography [97,99]	Can confirm adequate vascularization of a bowel segment May reduce bowel ischemic complications	Unclear whether it can effectively prevent anastomotic complications	Intravenous injection before bowel resection and after the completion of the anastomosis
	Ureteral vascularization assessment [98,99]	Confirm adequate vascularization May reduce post operative stricture rate	Unclear utility in centers with already low incidence rates of strictures	Intravenous injection prior to anastomosis
	Lymphangiography and sentinel node biopsy [94,96]	May improve staging by targeted removal of LNs outside the standard ePLND template	Only exploratory reports Low specificity It cannot be used as an alternative to ePLND or in association with radiotracers High variable drainage related to tumor localization, size, type and extent of disease	Cystoscopic mucosal injection around the tumor
Ureteral reconstruction, Pyeloplasty	Ureteral vascularization assessment [87,88,89,90]	May predict and prevent postoperative obstruction and pyeloplasty failure Reduce unnecessary tissue disruption	Only reports with low sample size	Intravenous injection
	Identification of ureteral strictures, ureter and renal pelvis [87,88,89,90,100,101]	Simpler identification of anatomical structures in complex anatomy Facilitate better localization and delineation of stenosis’s characteristics Ensure quality control after anastomosis or reimplantation	Only reports with low sample size	Trans-nephrostomic injection Retrograde injection through ureteral catheter

## 10. Conclusions

This comprehensive and up-to-date literature review provides an in-depth analysis of the field of urological robot-assisted surgery, highlighting how technologies such as NIRF have fostered continuous advancements and the exploration of new techniques.

The application of NIRF extends to both oncological and non-oncological procedures, showcasing its versatility in robotic procedures, “stealing” from the past and reintroducing in the present, giving “new life” to well-established technologies.

The synergistic combination of intraoperative support, the safety profile, widespread availability and efficient cost management comprises an additional asset that empowers ICG-guided robotic surgery in the optimization of surgical techniques.

The use of Indocyanine Green as a contrast agent in NIRF offers both advantages and areas for improvement, paving the way for further research and refinement. Identifying these areas for enhancement allows for the exploration of new research avenues and the optimization of the effectiveness and applicability of ICG-NIRF in urology.

Although the analyzed studies may, in some cases, suffer from inadequate sample sizes, initial experiences and a lack of control groups, recurring motifs associated with the use of NIRF-guided surgery emerge: adjectives such as “easier,” “safer,” and “faster” are repeated in the discussions of the various cited articles.

Therefore, there may be an opportunity to evaluate ICG-guided surgery from another perspective: shifting the focus from the sole concept of oncological or functional benefit—which may not be consistently demonstrated using ICG—towards an academic purpose. NIRF surgery could not only offer experienced surgeons the opportunity to tackle complex interventions, but could also enable more effective mentoring for young surgeons. By leveraging the advantages provided by this technique in identifying anatomical structures and guiding surgical dissection, the possibility of safely approaching various surgical urological techniques becomes evident.

In conclusion, this review serves as an important resource for understanding the current state of the art of NIRF-based urological surgery.

## Figures and Tables

**Figure 1 sensors-23-05497-f001:**
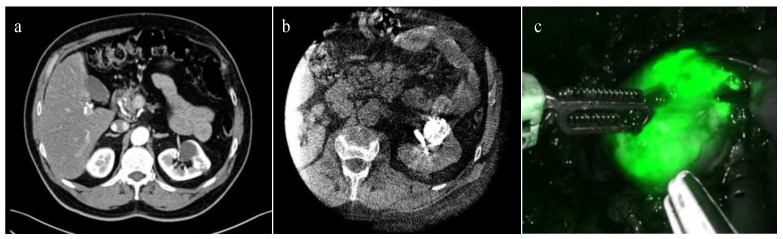
ICG-guided Management of Endophytic Renal Tumors: (**a**) CT-scan showing a totally endophytic left renal tumor; (**b**) trans-arterial super selective embolization “tagging” with ICG-Lipiodol; and (**c**) renal tumor intraoperative ICG near-infrared imaging identification.

**Figure 2 sensors-23-05497-f002:**
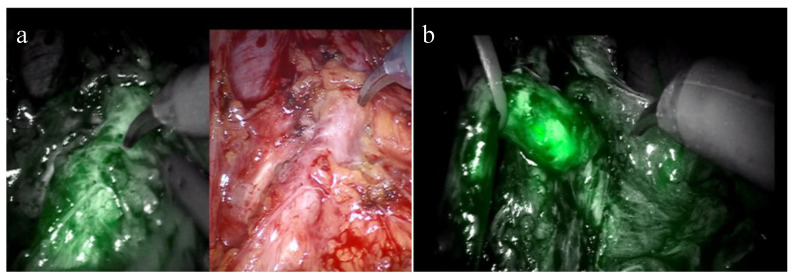
Trans-nephrostomic Indocyanine-Green-guided Robotic Ureteral Reimplantation for Benign Uretero-Ileal Strictures after Robotic Cystectomy: (**a**) Alternate use of conventional imaging and NIRF for lumbar ureter identification; (**b**) NIRF helping in the identification of the ureteral–neobladder junction stricture.

## Data Availability

No new data were created during this study.

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
