# Peer review of "Exploring the Applications of Indocyanine Green in Robot-Assisted Urological Surgery: A Comprehensive Review of Fluorescence-Guided Techniques"

_sensors, 2023, doi:10.3390/s23125497_

Round 1

Reviewer 1 Report

The article "Exploring the Applications of Indocyanine Green in Robot-Assisted Urological Surgery: A Comprehensive Review of Fluorescence-Guided Techniques" is a well-structured, comprehensive exploration of a significant topic in urological surgery. It offers a wide-ranging review of ICG's practical applications in various surgical procedures and aptly illustrates the versatility of NIRF. While the article is dense with information, this detail is necessary given the complexity of the topic. However, there is room for improvement in terms of presenting a more balanced perspective on the limitations or potential disadvantages of the technologies discussed. It is recommended to include more recent studies or data to solidify the claims. Overall, despite the minor areas for improvement, the article contributes significantly to the field and should be considered for publication. It could serve as a valuable resource for researchers, surgeons, and healthcare professionals interested in the current state of the art in NIRF-based urological surgery.

Major positive points in this review article are:

  1. The article is highly comprehensive, covering a vast array of topics related to the use of Indocyanine Green (ICG) in robot-assisted urological surgery.

  2. The exploration of the uses of ICG across different surgical procedures (renal, adrenal, prostate, penile, upper urinary tract, and bladder surgeries) provides an in-depth overview of its practical applications.

  3. The division of sections based on different surgical procedures makes the information organized and easy to navigate for readers.

  4. The article aptly highlights the versatility of Near-Infrared Fluorescence (NIRF), discussing its applicability in both oncological and non-oncological procedures.

  5. It details the benefits of ICG-guided robotic surgery, such as intraoperative support, safety profile, and cost-effectiveness, while also acknowledging areas for improvement.

Reviewer 2 Report

The study is aimed to review the current state of the art and illustrate the potential advantages and broad applications of combining indocyanine green-fluorescence guidance with robotic-assisted urological surgery.  The title is “Exploring the Applications of Indocyanine Green in Robot-Assisted Urological Surgery: A Comprehensive Review of Fluorescence-Guided Techniques”.

1.        This is a review article.

2.        Please also summarize in the form of a “Table”.

3.        Please review the literature and add more details in the discussion section.

4.        What is the new knowledge of the report?

5.        Please recommend “How to apply this knowledge?” to the readers.

 Minor editing of the English language required

Reviewer 3 Report

In this review, the authors review the potential advantages and broad applications of combining ICG fluorescence guidance with robot-assisted urological procedures. The relevant literature on various methods is fully analyzed and discussed.But there are some deficiencies that still need to be improved.

(1) Full text lacks images

(2) It is recommended that the author briefly summarize the advantages and disadvantages and technical prospects at the end of each type

Round 2

Reviewer 3 Report

The newly submitted manuscript is much improved and provides a comprehensive summary and insightful discussion. I suggest publishing this work in Sensors.